

# Genome-wide analysis of transcription factors related to anthocyanin biosynthesis in carmine radish (*Raphanus sativus* L.) fleshy roots

Jian Gao[1], Hua Peng[2], Fabo Chen[1], Mao Luo[3] and Wenbo Li[1]

[1] School of Advanced Agriculture and Bioengineering, Yangtze Normal University, Fuling, Chongqing, China
[2] College of Tourism and Cultural Industry, Sichuan Tourism College, Chengdu, Sichuan, China
[3] Drug Discovery Research Center, Southwest Medical University, Luzhou, Sichuan, China

Corresponding author
Wenbo Li, liverb@163.com

## ABSTRACT

Carmine radish produced in Chongqing is famous for containing a natural red pigment (red radish pigment). However, the anthocyanin biosynthesis transcriptome and the expression of anthocyanin biosynthesis-related genes in carmine radish have not been fully investigated. Uncovering the mechanism of anthocyanin biosynthesis in the 'Hongxin 1' carmine radish cultivar has become a dominant research topic in this field. In this study, a local carmine radish cultivar named 'Hongxin 1' containing a highly natural red pigment was used to analyze transcription factors (TFs) related to anthocyanin biosynthesis during the dynamic development of fleshy roots. Based on RNA sequencing data, a total of 1,747 TFs in 64 TF families were identified according to their DNA-binding domains. Of those, approximately 71 differentially expressed transcription factors (DETFs) were commonly detected in any one stage compared with roots in the seedling stage (SS_root). Moreover, 26 transcripts of DETFs targeted by 74 miRNAs belonging to 25 miRNA families were identified, including *MYB*, *WRKY*, *bHLH*, *ERF*, *GRAS*, *NF-YA*, *C2H2-Dof*, and *HD-ZIP*. Finally, eight DETF transcripts belonging to the *C2C2-Dof*, *bHLH* and *ERF* families and their eight corresponding miRNAs were selected for qRT-PCR to verify their functions related to anthocyanin biosynthesis during the development of carmine radish fleshy roots. Finally, we propose a putative miRNA-target regulatory model associated with anthocyanin biosynthesis in carmine radish. Our findings suggest that sucrose synthase might act as an important regulator to modulate anthocyanin biosynthesis in carmine radish by inducing several miRNAs (miR165a-5p, miR172b, miR827a, miR166g and miR1432-5p) targeting different ERFs than candidate miRNAs in the traditional WMBW complex in biological processes.

## INTRODUCTION

Radish (*Raphanus sativus* L.), a biennial root vegetable crop of the Brassicaceae family, is an economically important vegetable crop with an edible taproot. Currently, multiple

colors have been identified in the flesh of the taproot, including white, yellow, red, black and purple. The carmine radish 'Hongxin', produced in Chongqing, is famous for containing natural red pigment (red radish pigment). Anthocyanins are recognized as regulators of red to purple colors in nature and thereby produce water-soluble pigments belonging to the flavonoid group (*Khoo et al., 2017*). Most research has demonstrated that anthocyanins, a beneficial food additive worldwide, could pose major public health threats (cardiovascular disease, inflammation, obesity, and diabetes) caused by the chemical synthesis of food additives (*He & Giusti, 2010*; *Yousuf et al., 2016*). In addition, most regulatory genes have been extensively found to be involved in the anthocyanin biosynthesis pathway, which is largely conserved among flowering plants (*Bajpai et al., 2018*). Structural and regulatory genes could participate in anthocyanin biosynthesis. Previous studies have demonstrated that anthocyanins are first formed from phenylalanine by structural genes that encode a series of enzymes, such as phenylalanine ammonia-lyase (*PAL*), cinnamic 4-hydroxylase (*C4H*) and 4-coumarate-CoA ligase (*4CL*), followed by chalcone synthase (*CHS*), through phenylpropanoid metabolism. Then, the product 4, 2′ 4′ 6′-tetrahydrocychalcone is successively catalyzed by four enzymes (Chalcone Isomerase (*CHI*), flavanone 3-Hydroxylase (*F3H*), dihydroflavonol 4-Reductase (*DFR*), and anthocyanin synthase (*ANS/LDOX*)) (*Aza-González et al., 2013*; *Dao, Linthorst & Verpoorte, 2011*). Additionally, transcription factors (TFs) were demonstrated to be important regulatory genes involved in the anthocyanin biosynthetic pathway. TFs play vital roles in the normal development of an organism and in routine cellular functions (*Latchman, 1993*; *Yusuf et al., 2012*). Recently, researchers have demonstrated that MYB-bHLH-WDR (MBW) complexes could transcriptionally regulate genes encoding these enzymes through *MYB*, *bHLH* and *WD40* repeats (*Xu, Dubos & Lepiniec, 2015*). In addition, the regulatory genes Jasmonate zim-domain (*Qi et al., 2011*), Squamosa promoter binding protein-like (*Gou et al., 2011*) and *NAC* (*Zhou et al., 2015*) have also been reported to regulate anthocyanin biosynthesis. Moreover, recent evidence has shown that microRNAs (miRNAs) can act as important regulators of anthocyanin biosynthesis in plants (*Jia et al., 2015a*; *Yang et al., 2013*); examples include the miR156, miR165/166, miR828 and miR858 families in *Arabidopsis* (*Yang et al., 2013*) and *Solanum lycopersicum* (*Jia et al., 2015a*). However, the associations of these miRNAs with TFs have not been systematically investigated, and the related miRNAs involved in regulating anthocyanin biosynthesis have not been reported in carmine radish.

In this study, transcriptome analysis and functional validation of putative differentially expressed transcription factors (DETFs) involved in anthocyanin biosynthesis were performed using a local carmine radish cultivar named 'Hongxin 1', which contains a highly natural red pigment. As determined by RNA sequencing, a total of 1,747 TFs comprising 64 TF families were identified as being involved in the dynamic development of fleshy roots according to their DNA-binding domains. Of these TFs, approximately 71 DETFs were commonly differentially expressed in any one stage compared with SS_root. Of those 71 DETFs, 26 were predicted to be targets of 74 miRNAs belonging to 25 miRNA families, and many DETF gene products cleaved transcripts targeted by miRNAs,

including *MYB*, *WRKY*, *bHLH*, *ERF*, *GRAS*, *NF-YA*, *C2H2-Dof*, and *HD-ZIP*. In addition, eight DETF transcripts belonging to the *C2C2-Dof*, *bHLH* and *ERF* families and their eight corresponding miRNAs were selected for qRT-PCR to verify their functions in the development of fleshy roots in carmine radish. Finally, a putative miRNA-target regulatory model associated with anthocyanin biosynthesis in carmine radish was illustrated. Our findings suggest that miRNAs are involved in the regulation of anthocyanin biosynthesis in the dynamic development of fleshy roots in carmine radish.

# MATERIALS AND METHODS

## Quantification of anthocyanin levels in carmine radish

Five local cultivars of carmine radish ('Hongxin 1', 'Guanguan', 'Longquan 1', 'Yanzhi 1' and 'Yanzhi 2') containing natural red pigment (red radish pigment) collected from Fuling were selected as experimental materials. To identify the pigment contents of the five local cultivars of carmine radish, the dynamics of anthocyanin in the development of fleshy roots in carmine radish were investigated by HPLC analysis. The study groups included fleshy roots from the seedling stage (SS, 15 days after planting), initially expanded fleshy roots (IE, 40 days after planting), fully expanded fleshy roots (FE, 70 days after planting), fleshy roots from the bolting stage (BS, 120 days after planting), fleshly roots from the initial flowering stage (IFS, 140 days after planting), fleshy roots from the full-bloom stage (FBS, 160 days after planting), and fleshy roots from the podding stage (PS, 200 days after planting). Briefly, fleshy roots were collected from three homozygous individuals from the five local cultivars of carmine radish and pooled together; three replicates were included. The fleshy root tissues were ground in liquid nitrogen, and the red pigment was then extracted with a solvent mixture containing methanol (40%, v/v), formic acid (0.1%, v/v) and acetone (40%, v/v). We used a 10 μL injection volume on a VDS C-18 column (4.6 × 250 mm, 5 μm, VDS Optilab, Berlin, Germany) with a 0.8 mL min$^{-1}$ flow rate. Based on the results from HPLC analysis, RNA-Seq was used to investigate young fleshy roots in the carmine radish 'Hongxin 1' cultivar obtained from the dynamic fleshy root developmental stages, including fleshy roots from the SS, the initial expansion (IE) stage, the full expansion (FE) stage, the BS, the IFS, the FBS and the PS (Fig. S1). In this study, based on our private RNA sequencing data (PRJNA565866), transcriptome analysis and functional validation of putative DETFs related to the development of fleshy roots in 'Hongxin' carmine radish were conducted.

## Sample collection and RNA isolation

The carmine radish 'Hongxin 1' was cultivated in a greenhouse at the experimental farm of Yihe (Yangtze Normal University experiment base) in 2018. First, we sowed seeds of 'Hongxin 1' in sterilized soil for 2 weeks under normal growth conditions (23 °C, 16 h light/8 h dark). Next, 2-week-old plants were transferred and kept for 15 days in the cold room (5 ± 1 °C, 12 h light/12 h dark) for vernalization treatment. After the vernalization period, the plants were grown in a normal growth room under normal growth conditions (23 °C, 16 h light/8 h dark). At least three independent
biological replicates of fleshy roots obtained from the development stages of carmine radish 'Hongxin 1' were collected for qRT-PCR analysis of miRNAs and their related targets. All harvested tissues were immediately frozen in liquid nitrogen and stored at −80 °C for qRT-PCR. Subsequently, using the mirVana™ miRNA Isolation Kit (Ambion) and Trizol Reagent (Invitrogen, Nottingham, UK) kit according to the manufacturers' instructions, small and total RNAs were isolated from each sample.

## Gene annotation, enrichment analysis and cluster analysis of DETFs

Transcription factors were obtained from the available 'Hongxin' carmine radish transcription profile data. Subsequently, the TFs were subjected to GO annotation analysis using the Gene Ontology database (http://www.geneontology.org/) and to KEGG pathway enrichment analysis using KOBAS software (Xie et al., 2011). DETFs were then screened by noiseqbio (Tarazona et al., 2012) and identified using a corrected $P$-value < 0.05 between each set of compared samples (fold changes in the expression levels of genes in six radish cultivars, including 'IE_root', 'FE_root', 'BS_root', 'IFS_root', 'FBS_root' and 'PS_root', were identified by comparison with 'SS_root'—'IE_root' vs 'SS_root', 'FE_root' vs 'SS_root', 'BS_root' vs 'SS_root', 'IFS_root' vs 'SS_root', 'FBS_root' vs 'SS_root' and 'PS_root' vs 'SS_root'). DETFs were then clustered according to their expression patterns in the dynamic growth stages of carmine radish and plotted using the neighbor-joining cluster method through a homemade R script.

## Expression pattern analysis and regulatory pathway identification

To better understand the regulatory networks involving the DETFs, we searched putative DETFs targeted by miRNAs collected from the literature published by psRNATarget (Sun et al., 2015), and the parameters were set as follows: a maximum expectation of 3.5 and a target accessibility (UPE) of 50. Negative correlations between DETs and miRNAs were validated by quantitative real-time (Dai, Zhuang & Zhao, 2018) PCR.

## Validation of DETFs related to anthocyanin biosynthesis using qRT-PCR

To monitor all DETFs related to anthocyanin biosynthesis, eight differentially expressed transcription factors (DETFs, cluster_24061_AP2/ERF-ERF, cluster_24166_AP2/ERF-ERF, cluster_59539_GRAS, cluster_44972_B3, cluster_22927_NAC, cluster_10013_C2C2_Dof, cluster_5503_bHLH, cluster_12215_bHLH, cluster_8116_WRKY and cluster_16741_C3HH) were selected for qRT-PCR (ABI 7500 real-time PCR System, United States) based on their expression patterns (Table S1). In addition, eight DETFs related to miRNAs (cluster_10013_C2C2_Dof, cluster_23883_C2C2_Dof, cluster_17168_bHLH, cluster_16851_AP2/ERF-ERF, cluster_11321_AP2/ERF-ERF, cluster_19966_AP2/ERF-ERF, Cluster_6812_AP2/ERF-ERF, cluster_16203_AP2/ERF-ERF, and cluster_16851_AP2/ERF-ERF) were also selected for validation using qRT-PCR. The primers for the qRT-PCR experiments were designed

using Primer 5.0 software, and the radish actin gene was used as the reference gene (Table S2). First-strand cDNA synthesis was performed with 1 μg of total RNA from 'Hongxin 1' carmine radish using an M-MLV reverse transcriptase (Promega, Madison, WI, USA). The amplification programs were run according to the standard protocol of the ABI7500 system in triplicate as described by *Gao et al. (2015)*. The relative quantitative method ($2^{-\Delta\Delta CT}$) was used to calculate the fold changes in the expression levels of target genes (*Schefe et al., 2006*).

## Validation of the candidate miRNAs by qRT-PCR

To validate the putative miRNAs related to anthocyanin biosynthesis, eight miRNAs, including miR408a, miR172d, miR1432-5p, miR1425-5p, miR166g, miR827a, miR172b and miR165a-5p, were selected based on their corresponding DETFs. The primers for qRT-PCR experiments were designed using Primer 5.0 software, and the radish gene actin was used as a standard control (Table S2). First, using the One Step miRNA 1st cDNA Synthesis Kit (Shenggong, Chengdu, China), microRNA reverse transcription reactions were performed in an Eppendorf Mastercycler (Eppendorf North America, Westbury, NY, USA) for 60 min at 37 °C and for 5 min at 95 °C; the samples were then stored at 4 °C until further use. The RT-PCRs were performed in a 10 μL volume containing 1 μL of the diluted reverse transcription product, 1 μL of PCR buffer, 0.2 mM dNTPs, 2.0 U of EasyTaq DNA polymerase (TransGen Biotech, Beijing, China), 0.5 μM specific miRNA primer and 0.5 μM universal primer (5-TTACCTAGCGTATCGTTGAC-3) on an Eppendorf Mastercycler. The amplification programs were performed according to the standard protocol of the ABI7500 system in triplicate as described by *Gao et al. (2015)*. The relative quantitative method ($2^{-\Delta\Delta CT}$) was used to calculate the fold changes in the expression levels of target genes (*Schefe et al., 2006*).

# RESULTS

## Transcription factors related to anthocyanin biosynthesis in the development of fleshy roots of carmine radish: a summary

Based on GO and KEGG pathway analyses, TFs related to anthocyanin biosynthesis were identified in 'Hongxin 1' carmine radish. A total of 1,747 TFs in 64 TF families related to anthocyanin biosynthesis in the dynamic developmental stages of fleshy roots were obtained from RNA-Seq analysis based on their DNA-binding domains. Of these TF families, the *AP2/ERF-ERF*, *WRKY*, *NAC*, *bHLH*, *bZIP*, *MYB-related*, *C3H*, *C2H2*, *GARP-G2-like*, *MYB* and *GRAS* families were found to be the most represented, with 125, 114, 84, 80, 78, 73, 63, 55, 48, 48 and 43 members, respectively, and were expressed in at least one of the seven libraries (Fig. 1A).

The results showed that 1,682, 1,677, 1,568, 1,632, 1,633, 1,656 and 1,586 TFs related to anthocyanin biosynthesis were expressed in fleshy roots from the SS, the IE stage, the FE stage, the BS, the IFS, the FBS and the PS, respectively (Table S3). Of those, 1,415 (80.99%) TFs were expressed in each of the seven libraries. Of those, five candidate TFs belonging to four TF families (*AP2/ERF-ERF*, *GRAS*, *NAC* and *B3*) were found to exhibit opposite dynamic anthocyanin profiles in the fleshy roots of carmine radish, but five candidate TFs

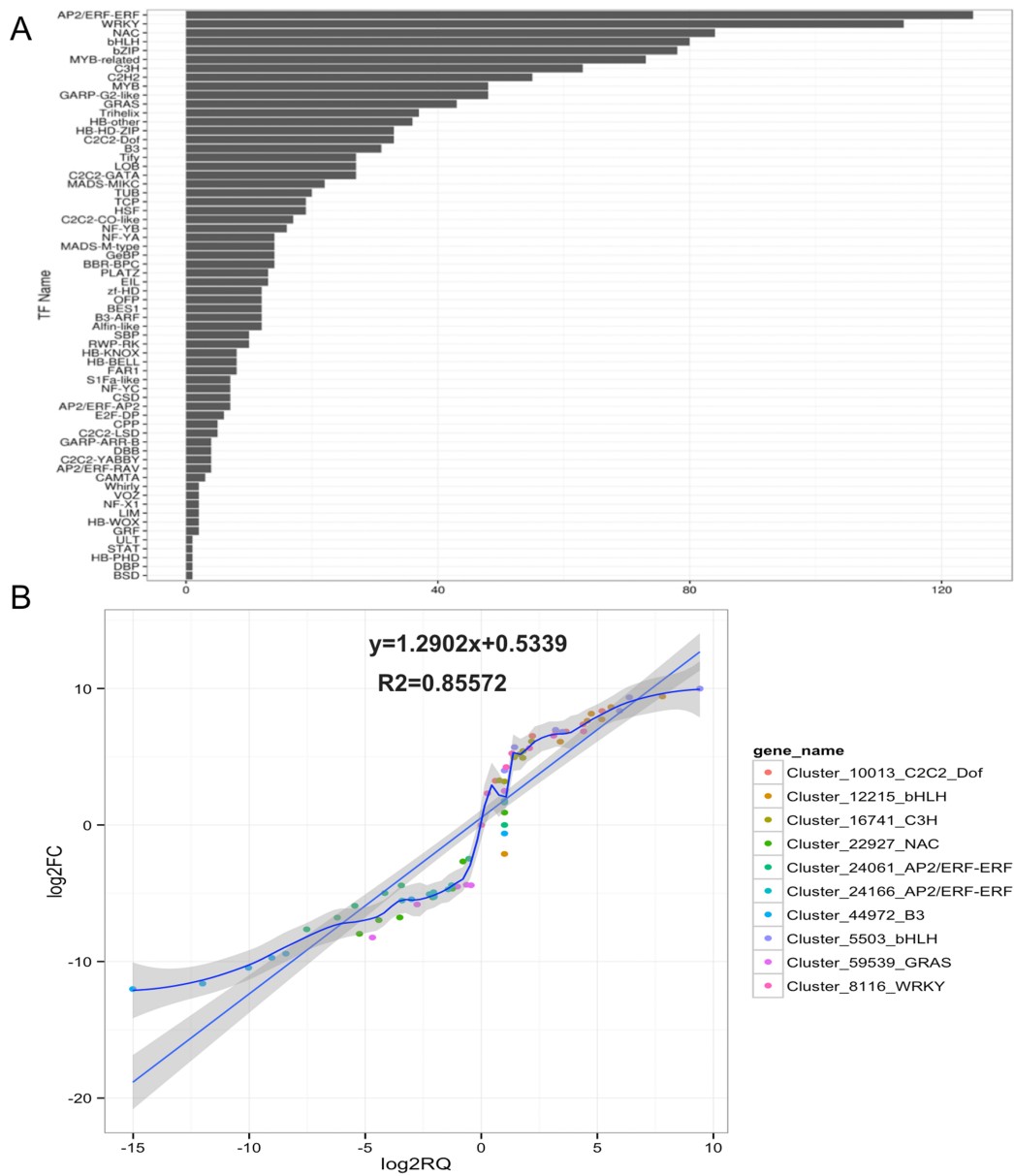

**Figure 1 Identification and qRT-PCR validation of *Transcription factors (TFs)* related to anthocyanin biosynthesis *in the development of fleshy roots in carmine radish*.** (A) Identification of transcription factors (TFs) related to anthocyanin biosynthesis involved in the development of fleshy roots in 'Hongxin 1' carmine radish (B) Validate of candidate TFs related to anthocyanin biosynthesis involved in the dynamics growing stages of fleshy roots in carmine radish using qRT-PCR and then correlation between RNA-seq and qPCR data were conducted. Each RNA-seq expression data was plotted against that from quantitative real-time PCR and fit into a linear regression. Both *x*- and *y*-axes were shown in log2 scale and each color represented a different gene.

belonging to four TF families (*C2C2-Dof, bHLH, WRKY* and *C3H*) were consistently found. Subsequently, those 10 putative TFs were selected and validated by qRT-PCR analysis. As shown in Fig. 1B, the results of qRT-PCR analysis were highly consistent with the expression profiles determined by RNASeq.

## Differentially expressed TFs (DETFs) related to anthocyanin biosynthesis involved in the development of fleshy roots in carmine radish

A total of 161 DETFs were identified to be related to anthocyanin biosynthesis during the development of fleshy roots in carmine radish when compared with the SS_root group ($P$ value <0.05, fold change <-1 or >1). In total, 116, 130, 136, 144, 141 and 136 DETFs were identified for 'IE_root_vs_SS_root', 'FE_root' vs 'SS_root', 'BS_root' vs 'SS_root', 'IFS_root' vs 'SS_root', 'FBS_root' vs 'SS_root' and 'PS_root' vs 'SS_root', respectively (Fig. 2A). In addition, these 161 DETFs were categorized into 43 families based on their DNA-binding domains (Table S4). Of those, the *AP2/ERF-ERF* (31 members), *WRKY* (nine members), *bHLH* (11 members), *NAC* (seven members), *MYB* (seven members) and *HB-HD-zip* (five members) families were found to be the most represented in the development of fleshy roots in carmine radish. Among the DETFs, approximately 71 were commonly differentially expressed in any one stage compared with the SS_root group using a venny graph (Fig. 2B). Subsequently, 161 DETFs were categorized into 4 distinct clusters using the K-means clustering algorithm (Fig. 2C). Among them, class I contained 24 substantially altered DETFs that were identified in the almost-whole stage and included the *AP2/ERF-ERF* (4 numbers), *AUX/IAA* (1 number), *bHLH* (1 number), *HD-ZIP* (3), *NAC* (2 numbers) and WRKY (four numbers) transcripts. A total of 45 genes belonging to class II showed slight dynamic changes during the development of fleshy roots.

The expression patterns of 67 TFs in class IV were found to oppose those in class I, as they were substantially changed in the almost-whole stage and included *AP2/ERF-ERF* (22 numbers), *bHLH* (six numbers), *MYB* (eight numbers) and *GRAS* (two numbers) transcripts. Class III contained 25 genes that did not show significant trends throughout the development of fleshy roots. We further performed KOBAS analysis to annotate the DETFs in classes I and IV that are involved in the development of fleshy roots (Fig. 2C).

## Validation of DETFs targeted by miRNAs related to anthocyanin biosynthesis in the development of fleshy roots in carmine radish

To investigate the functions of DETFs in carmine radish, the web-based program psRNATarget was selected to predict their related miRNAs. A total of 26 DETF transcripts, including *MYB*, *WRKY*, *bHLH*, *ERF*, *GRAS*, *NF-YA*, *C2H2-Dof*, and *HD-ZIP*, were predicted to be targets of 74 miRNAs belonging to 25 miRNA families (Table S5). In addition, we found that some miRNAs were associated with more than one DETF transcript. For example, miR395b could target *ERF* (Cluster_7379) and *WRKY* (Cluster_8116). In addition, some transcripts were regulated by more than one miRNA. For instance, ERF (Cluster_7379) was targeted by miR395a, miR395b and miR399b-5p, and WRKY (Cluster_8116) was targeted by miR319-5p, miR395b and miR535b (Table S5). Of those, eight DETF transcripts belonging to the *C2C2-Dof*, *bHLH* and *ERF* families and their eight corresponding miRNAs were selected for qRT-PCR to verify their functions in the development of fleshy roots in carmine radish (Fig. 3A). The expression levels of

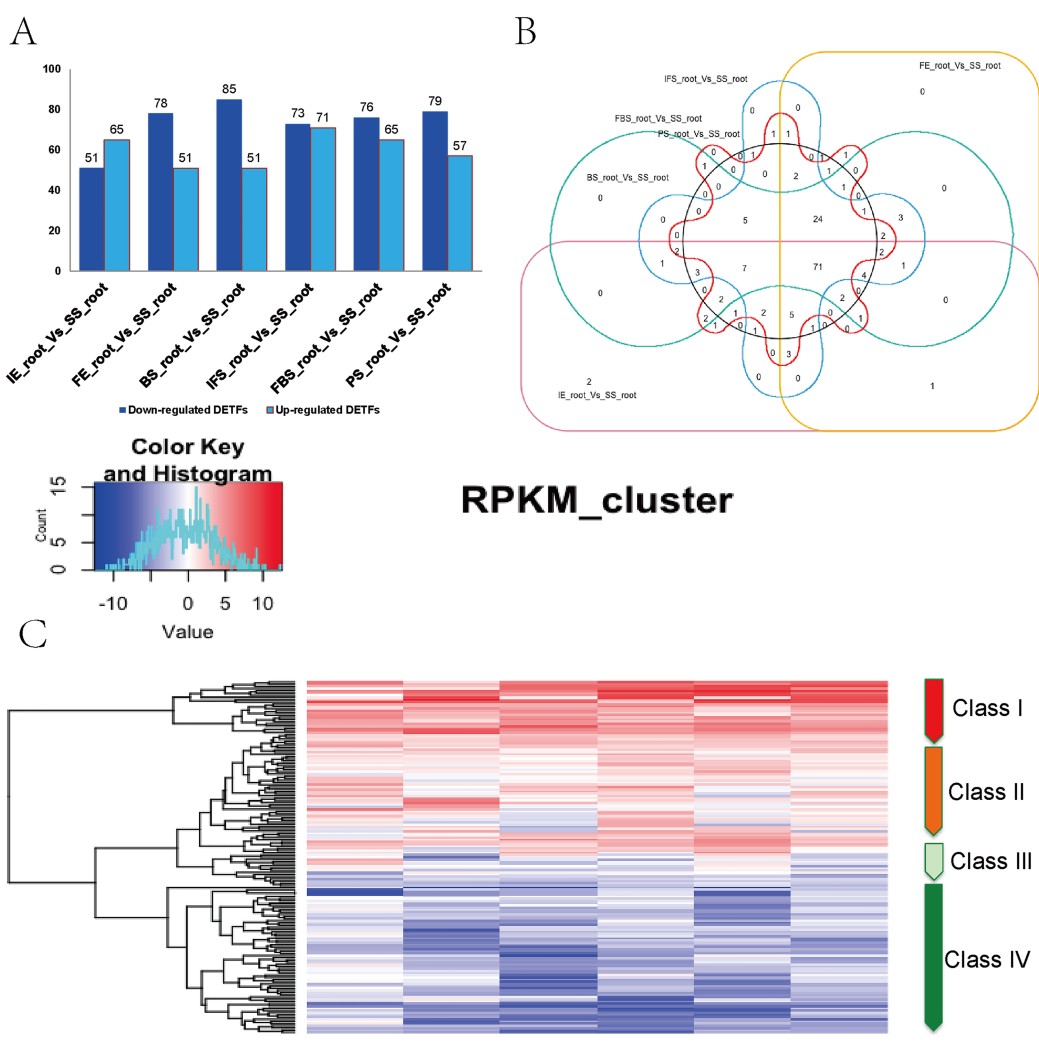

**Figure 2 Identification of DETFs related to anthocyanin biosynthesis and their expression patterns in the development of fleshy roots of carmine radish.** (A) Statistic of differentially expression transcription factors (including up-regulated and down-regulated in each comparison groups) related to anthocyanin biosynthesis in the dynamics growing stages of fleshy roots ('IE_root', 'FE_root', 'BS_root', 'IFS_root', 'FBS_root' and 'PS_root'), compared with 'SS_root' group. (B) Venny graph of co-modulated DEFTs (Common DETFs related to anthocyanin biosynthesis in carmine radish). (C) Clustering and heat map of common DETFs related to anthocyanin biosynthesis based on the expression profiles in the dynamics growing stages of fleshy roots ('IE_root', 'FE_root', 'BS_root', 'IFS_root', 'FBS_root' and 'PS_root'), compared with 'SS_root' group.

miR408a, miR1432-5p and miR166g were down-regulated in the development of fleshy roots in carmine radish 'Hongxin 1', whereas those of their corresponding target genes *C2C2-Dof*, *bHLH* and *ERF*, respectively, were upregulated in the 'Hongxin 1' cultivar (Figs. 3B, 3D, 3G). The expression levels of miR172b, miR165-5p, miR172d, miR827a and miR1425-5p were increased, while those of their corresponding target genes *C2C2-Dof* and *ERF*, respectively, were decreased in the 'Hongxin 1' cultivar (Figs. 3C, E, F, H and I). The results showed that inversely correlated expression patterns were found between the miRNAs and their corresponding targets.

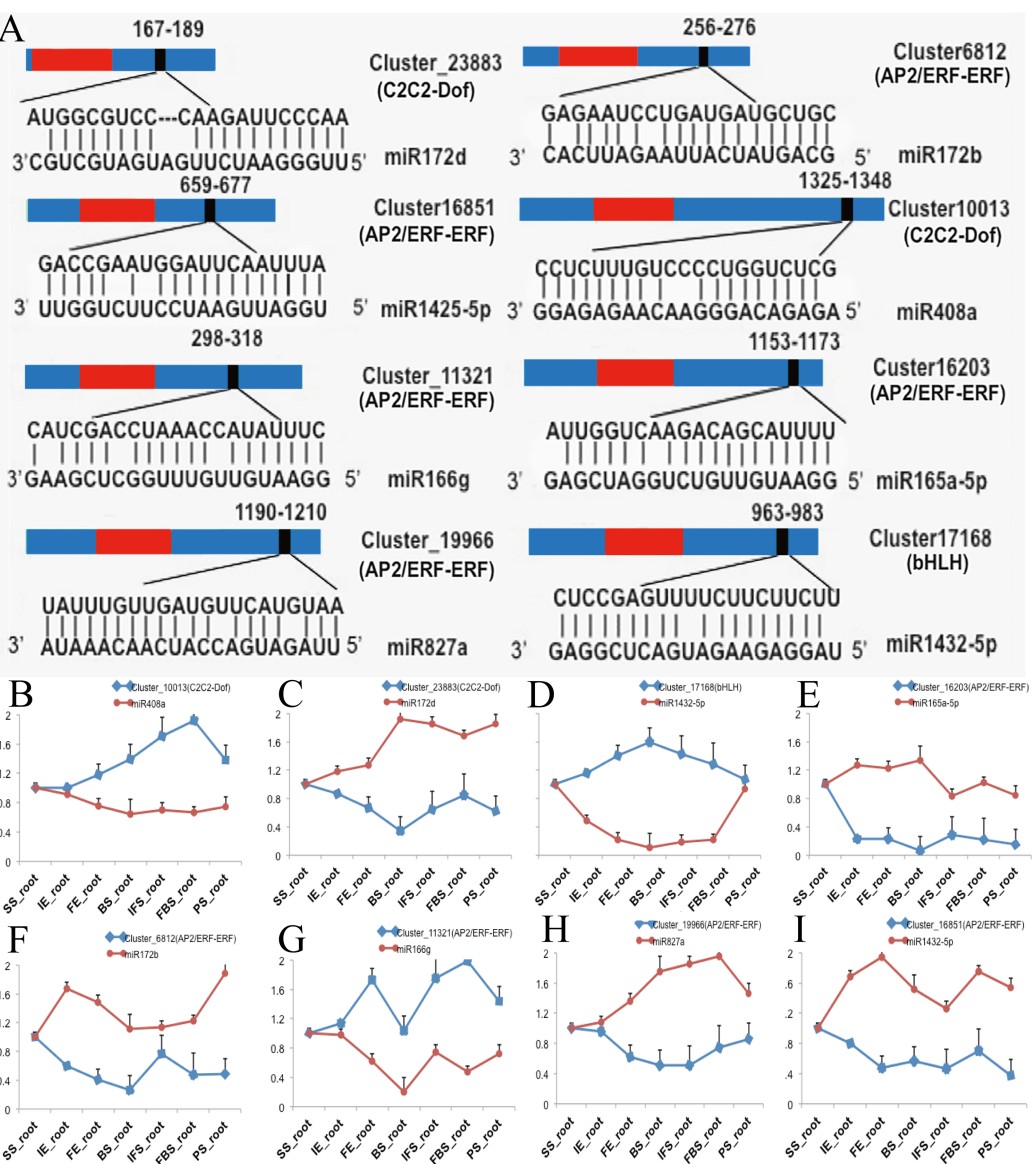

**Figure 3 Predict of miRNAs regulated by DEFTs related to anthocyanin biosynthesis, as well as validate the DETFs and their related miRNAs in the development of fleshy roots in carmine radish.** (A) Predict of miRNAs regulated by DEFTs using psRNATarget. The miRNA target sites (red) with the nucleotide positions of DEFTs transcripts are shown. The RNA sequence of each complementary site from 5′ to 3′ and the predicted miRNA sequence from 3′ to 5′ are indicated in the expanded regions. (B–I) qRT-PCR analysis of miRNAs and DETFs expressed in different samples in the development of fleshy roots in carmine radish. Standard error bars are provided for three biological repeats.

## Putative miRNA-target model associated with anthocyanin biosynthesis in carmine radish

Some TFs were shown to function as important regulators of anthocyanin biosynthesis, including *MYB*, *bHLH*, *WRKY*, *ERF* and *HD-Zip*. In addition, sucrose synthase, sugar/inositol transporter, and the WMBW complex were also found to potentially participate in

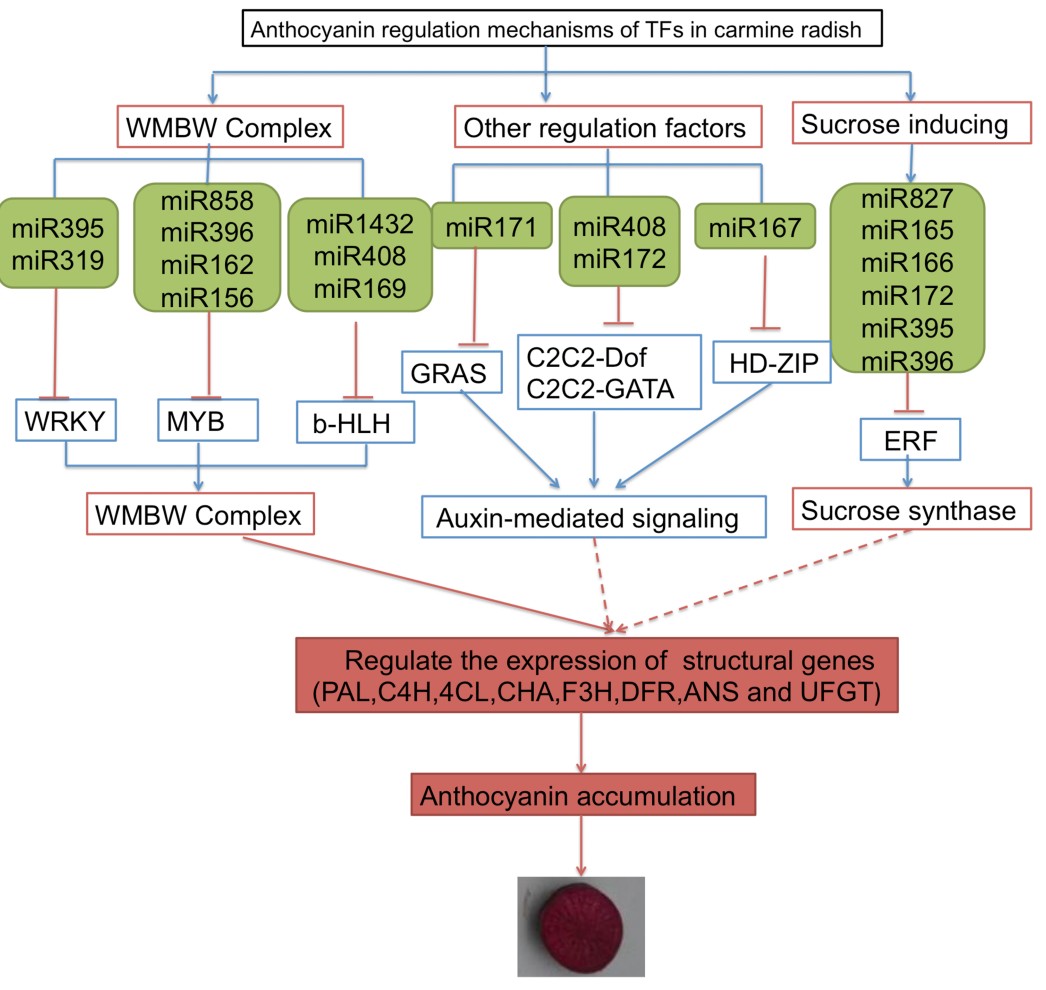

**Figure 4 A proposed model of transcription factor related to anthocyanin biosynthesis targeted by miRNAs were involved in the development of fleshy roots in carmine radish.** The genes in blue box denote DETs identified in the present research, as well as their responding miRNAs denoted in green box.

anthocyanin biosynthesis. Here, based on the annotations of DETFs, we identified 26 transcripts of DETFs targeted by 74 miRNAs belonging to 25 miRNA families that might participate in anthocyanin biosynthesis in carmine radish (Table S5). Here, we propose a putative miRNA-target model associated with anthocyanin biosynthesis in carmine radish (Fig. 4). The miR395, miR319, miR5720, miR1432 and miR408 families potentially targeted *WRKY*, *MYB* and *b-HLH* through the WMBW complex; the miR171, miR408 and miR172, miR167 families possibly targeted *GRAS*, *C2C2-Dof* and *HD-ZIP* to play roles in auxin-mediated signaling; and miR827, miR165/166, miR172 and miR395/miR396 targeted ERF to activate sucrose synthase. All three biological processes (involving the WMBW complex, auxin-mediated signaling and sucrose synthase) influence the expression of structural genes (*PAL*, *C4H*, *4CL*, *CHS*, *F3H*, *DFR*, *ANS* and *UFGT*) and subsequently modulate anthocyanin biosynthesis by forming a complex regulatory network.

## DISCUSSION

In plants, the MBW protein complex formed by the *R2R3-MYB*, *bHLH*, and *WD40* proteins could transcriptionally activate the promoters of structural genes related to anthocyanin synthesis (*Xu, Dubos & Lepiniec, 2015*). The complex is boosted by the participation of a WRKY transcription factor, which also confers specificity to other sets of target genes involved in, for example, vacuolar hyperacidification (*Verweij et al., 2016*). Previous studies showed that the expression of the MBW complex could be repressed by high levels of auxin to regulate the biosynthesis of anthocyanin (*Ji et al., 2015*; *Liu, Shi & Xie, 2014*). The *ARF* and *HD-ZIP* transcription factor genes in the *Aux/IAA-ARF* pathway were identified to regulate anthocyanin biosynthesis through auxin-mediated signaling in apples (*Wang, Hagen & Guilfoyle, 2013*). The *HD-ZIP* TF plays vital roles in regulating organ polarity in plants (*Williams et al., 2005*). In addition, TFs of the *GRAS* family have important control functions during primary and lateral root development, linking auxin signaling with cell specification and patterning, and are also involved in feedback regulation of local auxin homeostasis (*Benjamins & Scheres, 2008*). *C2H2-Dof* TFs are overrepresented in at least two auxin-inducible conditions in *Arabidopsis* (*Peng et al., 2009*). In this study, we demonstrated that *HD-ZIPs* and *GRAS* were targeted by miR167 and miR171, respectively, and that *C2C2-Dof* was targeted by miR408 and miR172; however, we did not identify a significant change in ARF in carmine radish. These results suggested that miR171, miR167, miR408 and miR172 might be involved in anthocyanin biosynthesis through auxin-mediated signaling.

The *MYB* TFs have been identified as important regulators of anthocyanin biosynthesis through the MBW complex (*Lai et al., 2014*; *Shen et al., 2014*). Anthocyanin production could be effectively induced by *AtPAP1* (*AtMYB75*) as well as by its orthologs in various plant species (*Rowan et al., 2009*; *Zuluaga et al., 2008*). Recently, *MYB* was identified as a target of many miRNAs in anthocyanin production in plants. An example is miR828 (*Guan et al., 2014*; *Xie, Sun & Dan-Ning, 2013*), which could cleave *MYBL2* to improve anthocyanin accumulation in *Arabidopsis* (*Wang et al., 2016*) and regulate two *SlMYB* transcripts for anthocyanin accumulation in tomato (*Jia et al., 2015b*). In this study, miR858, miR396, miR162 and ib-miR156 were identified to target *MYB* genes. In addition, a previous study showed that the transcription factor *bHLH* can activate the expression of *R2R3-MYB* and thereby activate or inhibit anthocyanin biosynthesis by binding to the promoters of *DFR* and *UFGT* (*Feller et al., 2011*; *Xie et al., 2012*). In this study, miR1432, miR408 and miR169 were identified to target *bHLH*.

In recent years, studies have demonstrated that sucrose synthase potentially participates in anthocyanin biosynthesis. A previous study showed that sucrose synthase could convert NDP-glucose and d-fructose into NDP and sucrose. Subsequently, the increased sucrose content caused significant anthocyanin accumulation (*Nagira et al., 2006*). Moreover, *Kühn et al. (2010)* demonstrated that miR854 could mediate the transport of sucrose through targeting the sugar/inositol transporter. In addition, more evidence indicates that the ethylene signaling pathway links sucrose signaling to anthocyanin accumulation (*Kwon et al., 2011*; *Harada et al., 2003*). Here, we found that nine miRNA

families targeting nine ERFs participate in anthocyanin biosynthesis through sucrose synthase, including miR172, miR166, miR399, miR1425, miR827, miR396, miR854, miR395 and miR863. Of those, 5 ERF DETFs and their related miRNAs were validated by qRT-PCR (Figs. 3B, C, 3E, 3F and 3G). Recently, Yao et al. (2017) reported that the ethylene response factor (ERF)/APETALA2 (AP2) TF *PyERF3* interacted with *PyMYB114* and its partner *PybHLH3* to coregulate anthocyanin biosynthesis in pears. We propose that ERFs might interact with the MBW protein complex through MYB and bHLH to coregulate anthocyanin biosynthesis in carmine radish, and further investigation of the MYB interaction with ERF to coordinately regulate fruit anthocyanin biosynthesis in carmine radish is needed.

In this study, we propose a putative miRNA-target model associated with anthocyanin biosynthesis in carmine radish (Fig. 4). The *miR395*, *miR319*, *miR5720*, *miR1432* and *miR408* miRNA families potentially targeted *WRKY*, *MYB* and *b-HLH* through the WMBW complex; the miR171, miR408 and miR172, miR167 families possibly targeted *GRAS*, *C2C2-Dof* and *HD-ZIP* to play roles in auxin-mediated signaling; and miR827, miR165/166, miR172 and miR395/miR396 targeted ERF to activate sucrose synthase. All three biological processes (involving the WMBW complex, auxin-mediated signaling and sucrose synthase) influence the expression of structural genes (*PAL*, *C4H*, *4CL*, *CHS*, *F3H*, *DFR*, *ANS* and *UFGT*) and subsequently modulate anthocyanin biosynthesis by forming a complex regulatory network. More interestingly, more *ERF* DETFs than *WRKY*, *MYB* and *bHLH* DETFs were identified in this study. We proposed that sucrose synthase might be an important biological agent that influences the expression of structural genes (*PAL*, *C4H*, *4CL*, *CHS*, *F3H*, *DFR*, *ANS* and *UFGT*), thereby modulating anthocyanin biosynthesis in carmine radish; however, the mechanism by which ERFs modulate anthocyanin biosynthesis through sucrose synthase needs further investigation.

## CONCLUSION

Here, 1,747 TFs belong to 64 TFs families were identified in a highly natural red pigment contained local cultivar named 'Hongxin 1', according to their DNA-binding domains. Of those, 26 transcripts of DETFs targeted by 74 miRNAs belong to 25 miRNAs family were identified, such as *MYB*, *WRKY*, *bHLH*, *ERF*, *GRAS*, *NF-YA*, *C2H2-Dof*, *HD-ZIP*, etc. Finally, eight DETFs transcripts belong to *C2C2-Dof*, *bHLH* and *ERF* family and their eight corresponding miRNAs were selected for qRT-PCR to verify their function related to anthocyanin biosynthesis. Our findings proposed that sucrose synthase might act as important regulators to modulate anthocyanin biosynthesis in carmine radish, through induce several miRNAs (miR165a-5p, miR172b, miR827a, miR166g and miR1432-5p) targeted different ERFs, compared with traditional WMBW complex biological process.

### Funding

This work was supported by the Natural Science Foundation of CSTB (cstc2019jcyj-msxm1573) and the Science and Technology Plan Projects of Fuling District

(FLKJ, 2018BBB3009). The funders had no role in study design, data collection and analysis, decision to publish, or preparation of the manuscript.

### Grant Disclosures

The following grant information was disclosed by the authors:
Natural Science Foundation of CSTB: cstc2019jcyj-msxm1573.
Science and Technology Plan Projects of Fuling District: FLKJ, 2018BBB3009.

### Competing Interests

The authors declare that they have no competing interest.

### Author Contributions

- Jian Gao conceived and designed the experiments, performed the experiments, prepared figures and/or tables, authored or reviewed drafts of the paper, approved the final draft.
- Hua Peng performed the experiments, analyzed the data, contributed reagents/materials/analysis tools, prepared figures and/or tables, approved the final draft.
- Fabo Chen performed the experiments, contributed reagents/materials/analysis tools, prepared figures and/or tables, approved the final draft.
- Mao Luo analyzed the data, prepared figures and/or tables, approved the final draft.
- Wenbo Li conceived and designed the experiments, prepared figures and/or tables, authored or reviewed drafts of the paper, approved the final draft.

### Data Availability

   All the raw read sequences are available at NCBI: PRJNA565866.

### Supplemental Information

Supplemental information for this article can be found online at http://dx.doi.org/10.7717/peerj.8041#supplemental-information.

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
