# Peer review of "Genome-wide analysis of transcription factors related to anthocyanin biosynthesis in carmine radish (Raphanus sativus L.) fleshy roots"

_PeerJ, doi:10.7717/peerj.8041_

## Round 0.1 · original submission · Major Revisions

The manuscript got three critical reviews demanding serious revision. All the reviewers ask you to check the English. The topic of this work is interesting and should be presented in a publication. It is necessary to study anthocyanin biosynthesis in related plant species. So, I recommend you use all the remarks and revise the manuscript thoroughly.

Reviewer 1 ·

Basic reporting

1、The language should be improved by English expert.
2、I concern where the transcriptome data come from. Had these data been deposited in the public database?
3、The authors did not describe the data of miRNAs used in this paper clearly.

Experimental design

In my opinion, the transcriptme data used for the following analysis should be with at least three biological replicates. please describe this part in detail.

Validity of the findings

The authors had described the expression profiles of somes miRNAs and their corresponding target genes. Did these miRNAs cleavge their target genes? could you validate this conclusion by 5' RLM-RACE?

Additional comments

1 、The language should be improved by English expert.
2、The title does not conform to the content of the paper exactly, which focused on the biosynthesis of anthocyanin.
3、Please use the abbreviations regularly. When the item appeared in the paper firstly, please use the style of full name (abbreviation). Then use the abbreviation after that.
4、Could you validate the target of miRNAd by 5' RLM-RACE?

Reviewer 2 ·

Basic reporting

BASIC REPORTING
In this manuscript, authors present a research about the identification of transcription factor families during the development of carmine radish (Raphanus sativus) roots. Besides, they show results about the expression miRNAs and their targets by RT-qPCR. They also relate the expression of TFs genes and miRNAs with the anthocyanin biosynthesis in roots. Finally, a model of the anthocyanin biosynthesis regulation mediated by miRNAs and their respective TFs targets is proposed. This research could be a good start point to understanding the anthocyanin biosynthesis in roots of carmine radish.
1. The most important point of this manuscript is the focus and the obtained results. Authors should not confuse the development of the roots with the anthocyanin biosynthesis. The development concept is a broad term involving growth of root structures, between other physiological changes, while the anthocyanin biosynthesis is product of the secondary metabolism. I suggest relating the anthocyanin biosynthesis, their TFs and miRNAs related the most important topic, and other TFs without scientific evidence in the regulation of the flavonoids could be omitted. Authors should improve the text writing of this point in the introduction, results and discussion.

2. A deep review of the English language is required.

3. Some figures could be moved to supplementary information (pdf review).

4. Bibliography: some cites into the text and the list references should be reviewed.

Experimental design

1. Question about the target genes: TFs constituting the MBW complex (MYB, bHLH and WD40) are the most important for the regulation of the anthocyanin biosynthesis, so, the authors should justify why the ERF and C2C2-Dof families were used for search of miRNA regulators.
2. Material and Methods section:
2.1. Information about the methods is lacking. For instance, more information is necessary to understand the expression analysis by RT-qPCR about the miRNAs. Please add all information.
2.2. Some subheadings is not related with the description the method. For example: “Expression pattern analysis and regulatory pathway identification”.
2.3. Information about the methods of anthocyanin levels quantification (Fig. S1) should be described.
2.4. If you have data about the statistical significance, you should include it. Please, also to include number of biological and/or technical replicates for anthocyanin quantification and RT-qPCR analysis.

Validity of the findings

1. Statistical analysis should be showed for the expression analysis.
2. Discussion should be improved. I recommend to focus in the possible implications of target TFs by miRNAs, and how it is related with the anthocyanin levels.
3. Please, to add some general conclusions.
4. The final model (Fig. 4) should be modified, considering that many experiments are necessary for clarify that the TFs that you are reporting are involved in the anthocyanin biosynthesis. Besides, you consider that some TFs that regulate organ polarity also regulate the anthocyanin biosynthesis.

Additional comments

Dear Gao et al.,

I read and and reviewed the manuscript entitled "Genome-wide analysis of transcription factors involved in development of carmine radish (Raphanus sativus L.) fleshy roots". It is a interesting research about the TFs and miRNAs regulating the anthocyanin biosynthesis, however, you should improve the focus, the methodology and methods information and the discussion. You can review these items in general comments and the attached pdf.

Best regards.

Annotated reviews are not available for download in order to protect the identity of reviewers who chose to remain anonymous.

Reviewer 3 ·

Basic reporting

The English should be improved in the entire manuscript, and there are some typographical errors in this article. For example, line 106, 'six cultivars',line 195, 'hongxin1' should be 'Hongxin1'and line 245, the citation 'Yulong et al.2016' should not be italic.

Experimental design

The first part of 'Materials and methods' section is 'Experiment design and RNA isolation' ,however, the authors did not describe how to isolate the RNA in this part.

Validity of the findings

The author did not state the replication of the experiment.

Additional comments

In this study, the authors performed a genome-wide analysis of transcription factors during carmine radish development based on the transcriptom data and proposed a putative miRNA-target model in anthocyanin biosynthesis. However, the conclusion is stated according to the transcriptom data, online miRNAs and targets prediction and a little qRT-PCR analysis, lacking the solid experimental support. Otherwise, the development of carmine radish is a complex process including transformation of nutrition, cell expansion, secondary metabolism except for anthocyanin biosynthesis, etc.. Some TFs identified in this article such as WRKYs and ERFs may mainly participate in other process rather than anthocyanin biosynthesis. So, it is too arbitrary to say these TFs modulate anthocyanin biosynthesis when there is no direct empirical evidence support it.

---

## Round 0.2 · Minor Revisions

There are some critical comments and suggestion. Please use it to update the text. Pay attention to the English presentation. I recommend ask native English speaker. Waiting your revision and final manuscript submission.

Reviewer 2 ·

Basic reporting

Authors considered the comments suggested in the first manuscript revision: Question 1, Q3, Q5-6, Q8, Q10, Q11 (according to ‘Response to reviewers’ file) are correct and well explained. However, I suggest that responses of Q4, Q7, Q9 and Q12 should be improved. Next, I indicate the main changes and several details in the text:

Q2. A deep review of the English language is required.
Response: We thank the reviewer for making this comment. We have polished the English of our manuscript carefully.
Reviewer response: Please, Could you do the next changes?
• Line 36: to remove “a”.
• Line 53: “layse” by “lyase”.
• Line 63-64: “recent researcher” by “Recently, researches…”
• Line 69: change “which” by “these miRNAs have not been…” or similar.
• Line 84: “Total” in lower case.
• Line 103: Please, change “dynamics Anthocyanidin profiles” by “anthocyanin profile dynamics” or something like that.
• Lina 139: “were identified obtained” by “were obtained”.
• Line 172: “are designed” by “were designed”.
• Line 173: “radish gene (Actin)” by “radish actin gene”. “standard control” by “reference gene” or “housekeeping gene”.
• Line 210: “…43 members and…” by “43 members, respectively; and…”.
• Line 211: to remove “respectively”.
• Line 266: Please, to modify “were identified increased” by “were increased”, “increased” or something like that.

Q4. Bibliography: some cites into the text and the list references should be reviewed.
Response: Thanks for your suggestions, we have checked all cites in the list references and checked them thoroughly, and fixed them carefully following the queries of PEERJ article.
Reviewer response: Some additional references should be improved.
• Line 56: ‘Anju Bajpai et al. 2017’. This cite is wrong wrote. Besides, the corresponding reference in the bibliography list has different write format of the others.
• Line 195: Please, change it “Jian et al.(Jian et al. 2015)” by “Jian et al. (2015)”.
• Line 300: ‘(WILLIAMS 2005)’. Please, change it by “(Williams 2005)”.
• Line 328-329: “Kuhn et. al” and “(Kühn & Grof 2010)” are write differently.
• Line 335-336: I think that “Gaifeng Yao et al.” should be written like “Yao et al.”


New comments for “BASIC REPORTING”:
• Please, write Raphanus sativus in italics.
- Lines 32, 88-89, 253, 277, 365: Please, change “25 miRNAs family” by “25 miRNA families”.
• Line 326: “D-fructose” by “D-fructose”
• Line 189: change “RT-PCR” by “qRT-PCR”.
• Please, change “anthocyanidin” by “anthocyanin” through the text.

Experimental design

Q7. Some subheadings is not related with the description the method. For example: “Expression pattern analysis and regulatory pathway identification”.
Response: Thanks for your good suggestions; we have fixed all the subheadings in the new modified.
Reviewer reponse: Thanks for consider this suggestion. I think that the new subheading ‘experiment design’ should be changed to “Quantification of anthocyanin levels” or similar.

Q9. If you have data about the statistical significance, you should include it. Please, also to include number of biological and/or technical replicates for anthocyanin quantification and RT-qPCR analysis.
Response: Thanks a lot, we have included the number of biological and/or technical replicates for anthocyanin quantification and RT-qPCR analysis in the new manuscript.
Reviewer reponse: Thanks. However, I can not find the number of replicates used for anthocyanin quantification. Please, add it.

Validity of the findings

Q12. Please, to add some general conclusions.
Response: Thanks a lot, we have added general conclusions in the new manuscript.
Reviewer reponse: I think that ‘conclusiones’ section can not be a copy of the abstract. Could you change it?


New comments for “VALIDITY OF THE FINDINGS”:
1. Line 161: You indicate that “ten DETFs were used for qRT-PCR analysis. However, only 8 DETFs are showed in Figure 3. Please, explain this or change “ten” for “eight”.
2. Line 280: miRNA5720 can not find in the proposed model.

Additional comments

Dear authors,

I reviewed the new manuscript version. I think that the article is better understood with your changes. But, I think that you should consider some minor changes.

Best regards.

---

## Round 0.3 · accepted · Accept

Thank you for the update, as I see all the remaining minor remarks were taken into account. Additional comment was on English language. Please ask PeerJ staff how to proceed. As academic editor I think the paper is acceptable now.